# Impact of emergency physician-staffed ambulances on preoperative time course and survival among injured patients requiring emergency surgery or transarterial embolization: A retrospective cohort study at a community emergency department in Japan

Yuko Ono [1,2]*, Yudai Iwasaki[2,3], Takaki Hirano[2], Katsuhiko Hashimoto[2,4], Takeyasu Kakamu[5], Shigeaki Inoue[1], Joji Kotani[1], Kazuaki Shinohara[2]

1 Department of Disaster and Emergency Medicine, Graduate School of Medicine, Kobe University, Kobe, Japan, 2 Department of Anesthesiology, Ohta General Hospital Foundation, Ohta Nishinouchi Hospital, Koriyama, Japan, 3 Department of Anesthesiology and Perioperative Medicine, Tohoku University, Graduate School of Medicine, Sendai, Japan, 4 Department of Minimally Invasive Surgical and Medical Oncology, Fukushima Medical University, Fukushima, Japan, 5 Department of Hygiene and Preventive Medicine, School of Medicine, Fukushima Medical University, Fukushima, Japan

* windmill@people.kobe-u.ac.jp

## Abstract

Injured patients requiring definitive intervention, such as surgery or transarterial emboliza-tion (TAE), are an extremely time-sensitive population. The effect of an emergency physi-cian (EP) patient care delivery system in this important trauma subset remains unclear. We aimed to clarify whether the preoperative time course and mortality among injured patients differ between ambulances staffed by EPs and those staffed by emergency life-saving tech-nicians (ELST). This was a retrospective cohort study at a community emergency depart-ment (ED) in Japan. We included all injured patients requiring emergency surgery or TAE who were transported directly from the ED to the operating room from January 2002 to December 2019. The primary exposure was dispatch of an EP-staffed ambulance to the prehospital scene. The primary outcome measures were preoperative time course including prehospital length of stay (LOS), ED LOS, and total time to definitive intervention. The other outcome of interest was in-hospital mortality. One-to-one propensity score matching was performed to compare these outcomes between the groups. Of the 1,020 eligible patients, 353 (34.6%) were transported to the ED by an EP-staffed ambulance. In the propensity score-matched analysis with 295 pairs, the EP group showed a significant increase in median prehospital LOS (71.0 min vs. 41.0 min, P < 0.001) and total time to definitive inter-vention (189.0 min vs. 177.0 min, P = 0.002) in comparison with the ELST group. Con-versely, ED LOS was significantly shorter in the EP group than in the ELST group (120.0 min vs. 131.0 min, P = 0.043). There was no significant difference in mortality between the two groups (8.8% vs.9.8%, P = 0.671). At a community hospital in Japan, EP-staffed ambu-lances were found to be associated with prolonged prehospital time, delay in definitive

**Data Availability Statement:** All relevant data are within the manuscript and its Supporting Information files.

**Funding:** The authors received no specific funding for this work.

**Competing interests:** The authors have declared that no competing interests exist.

treatment, and did not improve survival among injured patients needing definitive hemostatic procedures compared with ELST-staffed ambulances.

## Introduction

Traumatic injury is a leading cause of death and disability, particularly among young people, and involves a substantial economic burden for society [1, 2]. For example, in the United States (US) alone, more than 50 million people are injured each year, resulting in approximately 169,000 annual deaths and a lifetime cost of USD 406 billion [1]. According to the World Health Organization, more than 5 million people die annually as a result of injuries, which accounts for 9% of the world's deaths [3]. To maximize the survival of injured patients, stabilizing circulation via early hemostasis is vitally important. Previous studies have shown that rapid patient transfer to a trauma center and rapid initiation of definitive care, such as surgery or transarterial embolization (TAE), strongly influence survival among severely injured patients [4–9]. The preoperative time course, such as prehospital length of stay (LOS), emergency department (ED) LOS, and total time to surgery or TAE, is therefore regarded as an important parameter in trauma care [10, 11].

Unlike prehospital medical providers in the US, Japanese emergency life-saving technicians (ELSTs), a first responder for patients with trauma, are not allowed to perform invasive procedures such as endotracheal intubation, cricothyroidotomy, or tube thoracostomy for living patients with trauma. Therefore, most Japanese tertiary medical centers, including the study site, run a hospital-based emergency physician (EP) delivery system for early initiation of advanced trauma care [12]. Nevertheless, the effect of EP deployment to the prehospital scene on outcomes among injured patients remains controversial: Several studies have shown that EP involvement in prehospital trauma care is associated with improved survival in patients with trauma [13–19] whereas other studies report no significant differences [20–23] or decreased survival [24]. Although injured patients requiring definitive intervention, such as surgery or TAE, are an extremely time-sensitive population, previous studies have not focused on this important trauma subset [13–24], and little is known regarding whether and how an EP delivery system affects the preoperative time course and patient survival. Additionally, despite hospital LOS being a sensitive indicator that reflects patients' disease burden, previous studies have not clarified how EP involvement in prehospital trauma care affects this important parameter, as compared with ELSTs [13–24]. Because running an EP-staffed ambulance requires considerable cost as well as human resources [12, 20], rigorous investigation of its effectiveness compared with that of a conventional ELST-staffed ambulance is warranted. In this study, we therefore aimed to clarify whether and how the preoperative time course and survival of injured patients requiring definitive intervention differ between EP-staffed ambulances and ELST-staffed ambulances.

## Methods

### Study design and setting

This was a retrospective cohort study at a community hospital located in a provincial Japanese city. The review board at Ohta Nishinouchi Hospital approved this study on 22 April 2021 (approval no. 39_2020). The committee waived the need for patient consent because the study was not randomized and assessed the clinical outcomes of routine practices. The hospital serves as a teaching and tertiary care facility for a population of 538,000 inhabitants within a

50-km radius. Annually, the ED receives > 5,500 ambulances and approximately 25% of these involve patients with trauma. The hospital has 30 emergency ward beds, 10 operating rooms, and 1 catheterization laboratory that is available for patients with trauma.

At the study site, injured patients are initially treated by a trauma resuscitation team that comprises attending EPs, emergency medicine residents, post-graduate year 1 or 2 junior residents, and nurses. At the request of EPs, trauma surgeons, anesthesiologists, and interventional radiologists immediately respond from within the hospital and are actively involved in trauma resuscitation during weekday business hours. If injured patients require emergency surgery or TAE on the weekend or during nighttime hours, these specialists respond from outside the hospital, typically within 1 hour.

### ELST-staffed ambulances

The ELST-staffed ambulance system in Japan has been described previously [25–27]. In brief, ambulance crews in Japan typically consist of three emergency medical service personnel, and at least one of whom is an ELST who has completed extensive training. Japanese ELSTs perform medical procedures at the scene less actively than paramedics in the US and Europe [26]. ELSTs in Japan are allowed to perform basic trauma life support procedures, including cervical collar application, pneumatic anti-shock garment application, backboard fixation, and manual bag ventilation but not invasive procedures including endotracheal intubation, cricothyroidotomy, thoracostomy, and pericardiocentesis for living patients with trauma.

### EP-staffed ambulances

Like most Japanese tertiary medical centers, the study hospital runs a prehospital emergency medical unit consisting of a trained ambulance driver, a nurse, an attending-level EP, and a resident; the unit is available 24 hours, 7 days a week. This hospital-based EP-delivery system is dispatched to the scene following a request by the regional medical control center or by ELSTs upon assessment of the patient at the scene. The ELST-staffed ambulance that responds first transports the injured patient to a location where an EP-staffed ambulance can safely join up. Then, the EP-staffed ambulance team provides advanced trauma care in the ambulance, such as ultrasound examination, intravenous and intraosseous fluid resuscitation, endotracheal intubation, cricothyroidotomy, tube thoracostomy, and pericardiocentesis. At the study site, resuscitative endovascular balloon occlusion of the aorta is usually inserted in the ED or catheterization laboratory under radiographic guidance, and no patients receive this intervention at the prehospital scene. After emergency care, the EP-staffed ambulance transports the patient to the ED of the study hospital.

### Participants and data sources

The current study comprised all injured patients requiring emergency surgery and who were transported directly from the ED to the operating room, as well as all patients with trauma requiring TAE who were transported directly from the ED to the catheterization laboratory between January 1, 2002 and December 31, 2019. The exclusion criteria were patients who received ongoing cardiopulmonary resuscitation at the initial contact and patients who were transported from other facilities. Injured patients who were transported by a helicopter emergency medical service were also excluded from the analysis to remove the influence of vehicle type in emergency services. The data were collected from a hospital-based electronic database, which prospectively captures each patient's age, sex, comorbidities, initial recorded vital signs, ED presentation date and time, preoperative time course including time from the emergency call to ED arrival (prehospital LOS), time from ED arrival to arrival in the operating room or

catheterization laboratory (ED LOS), total time to surgery or TAE (time from the emergency call to the arrival in the operating room or catheterization laboratory), hospital LOS (time from the hospital admission to the hospital discharge or transfer), and discharge disposition (home, secondary care hospital, and long-term care hospital). This database also records injury severity according to the Abbreviated Injury Scale (AIS) of each body region, the Injury Severity Score (ISS) [28], the Revised Trauma Score (RTS) [29], and the probability of survival; this is according to the Trauma and Injury Severity Scores method [30]. These parameters were entered into the database at the earliest possible opportunity by one of the authors (K.S.). To reduce the risk of biased assessment, the investigator who constructed the database (K.S.) did not participate in any of the statistical analyses. The minimal anonymized data set used in this study is available in the (S1 Data).

## Exposures and outcome measurement

The primary exposure was dispatch of an EP-staffed ambulance to the prehospital scene. The primary outcome measures were the preoperative time course including prehospital LOS, ED LOS, and total time to definitive intervention. The other outcome of interest in this study was in-hospital mortality. Differences in hospital LOS and discharge disposition among survivors were also compared between EP and ELST groups.

## Statistical analysis

All analyses were performed according to an a priori statistical analysis plan. Initially, both a crude and a matching analysis were performed between injured patients requiring emergency surgery or TAE between the ELST and EP groups. A matching analysis was designed using estimated propensity scores (PS) for each patient. To estimate the PS, a logistic regression model was fitted for EP-staffed ambulances as a function of patient demographics, including age, sex, Charlson Comorbidity Index [31, 32], presentation time and period (8:00–16:59, 17:00–23:59, and 24:00–7:59, weekend or weekday and 2002–2007, 2008–2013, and 2014–2019, respectively), initial recorded vital signs Glasgow Coma Scale (GCS) score, systolic blood pressure (SBP), and respiratory rate, trauma etiology (blunt or penetrating), injury distribution with AIS $\geq$ 3, and ISS. To optimize model fitting, patients' GCS score, SBP, and respiratory rate were categorized according to the scoring system of the RTS [29]. The Charlson Comorbidity Index was also divided into four groups (0, 1, 2, and $\geq$ 3). We calculated the $c$ statistic to evaluate the goodness of fit. The standardized difference (SD) was used to evaluate the covariate balance; an absolute SD of $>$ 10% represents meaningful imbalance [33]. Each patient with trauma who was transported to the ED by an ELST-staffed ambulance was matched with a patient who was transported to the ED by an EP-staffed ambulance, with the nearest estimated propensity on the logit scale within a specified range (0.2 of the pooled standard deviation of estimated logits). If two or more patients in the ELST group met this criterion, one patient was randomly selected for matching. The Mann–Whitney U test was used to compare the preoperative time course and hospital LOS between the ELST and EP groups. Chi-squared tests were used to compare hospital mortality between the two groups. Differences in disposition locations between groups were assessed using a chi-squared test followed by residual analysis. All statistical analyses were performed using SPSS Statistics for Windows, version 25.0 (IBM Corp., Armonk, NY, USA). A p value of $<$ 0.05 was considered to indicate statistical significance.

## Subanalysis

To evaluate the robustness of the PS matching analysis described above, inverse probability of treatment weighting (IPTW) analysis was also conducted for hospital mortality. With the full

cohort, an unconditional logistic regression model adjusted for PS was also fitted using hospital mortality as a dependent variable. This logistic regression analysis was repeated in the subgroup of injured patients age > 55 years, ISS > 15, SBP < 90 mmHg, GCS score < 9, probability of survival < 0.5, torso injury requiring laparotomy or thoracotomy, and TAE and in the subgroup of patients presenting during off-hours (17:00 to 7:59 on weekdays plus all weekend hours). This approach was selected owing to the small sample size of each subgroup and the potential loss of statistical power using the PS-matching method.

Time from the emergency call to ED arrival and time from ED arrival to arrival in the operating room or catheterization laboratory were also compared by drawing cumulative event rate curves and using log-rank tests. To assess the independent effect of EP-staffed ambulance dispatch on prehospital LOS, ED LOS, and total time to the operating room or catheterization laboratory, we used Cox proportional hazards regression analysis for time-to-event data. By referencing previous relevant studies [15, 20, 21], age, sex, ISS, initial documented vital signs (GCS score, SBP, and respiratory rate), and trauma etiology (blunt or penetrating) were selected as covariates. To determine factors associated with delayed definitive intervention (time from an emergency call to surgery or TAE > 180 min), we evaluated differences in the characteristics between patients who received surgery or TAE within 180 min and those who received these procedures later. The cumulative event rate curves were generated using Graph-Pad Prism 8 (GraphPad Software, San Diego, CA, USA). We also evaluated whether distributions of time to death differed between the EP group and ELST group.

## Power analysis

The retrospective nature of the study predetermined the sample size. Therefore, the estimation of statistical power in advance was not possible. The observed power was computed post hoc using G*Power 3 for Windows (Heinrich Heine University, Dusseldorf, Germany).

## Results

### Participant flow

During the study period, 24,471 patients with trauma were transported to the ED, of whom 1,911 (7.2%) required emergency trauma surgery or TAE (Fig 1). Among them, we excluded 627 patients who were transported from other facilities and 264 patients who were transported by a helicopter emergency medical service. The remaining 1,020 patients were included in the crude analysis. Using one-to-one PS matching, we selected 295 pairs of patients who were transported by EP-staffed ambulance or by ELST-staffed ambulance. Complete records were available for all patients, and no data were missing from the analyses. The *c*-statistic for goodness of fit was 0.74 (95% confidence interval [CI] 0.71–0.77) in the PS model. S1 Fig shows the distributions of PS in the full and matched cohorts.

### Characteristics of study participants

Table 1 shows the demographics of all patients (n = 1,020) and PS-matched patients (n = 590). We observed statistically significant differences in vital signs such as GCS score, SBP, and respiratory rate between the two groups: All of these physiological parameters were severer in the EP group than in the ELST group. Similarly, patients in the EP group had more severe injuries, with higher ISS and a higher proportion with AIS ≥ 3 of the head or neck, chest, abdomen or pelvic contents, and extremities or pelvic girdle than patients in the ELST group. Patients transported by EP-staffed ambulance were more likely to receive TAE, laparotomy, or thoracotomy. As compared with ELST-staffed ambulances, EP-staffed ambulances were more likely

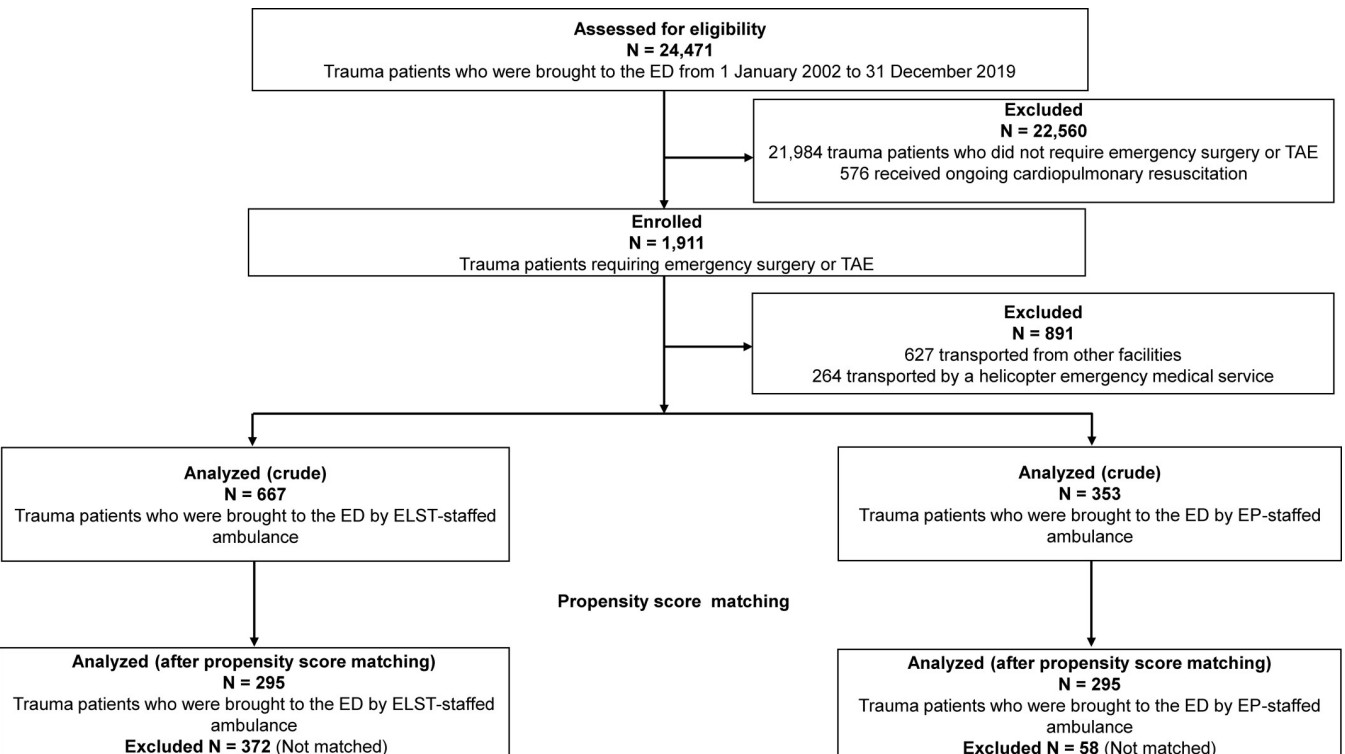

**Fig 1. Flow chart showing the selection process for injured patients included in the analyses.** ED, emergency department; ELST, emergency life-saving technician; EP, emergency physician; TAE, transcatheter arterial embolization.

to be dispatched during 2008–2013 and 2014–2019, between 8:00–16:59, and on weekdays. After PS matching, patient distributions were closely balanced, with all SD < 10% between the two groups.

S1 Table presents a comparison of pre-hospital treatment. Of 353 injured patients transported by EP-staffed ambulance, 40 (11.3%) received endotracheal intubation, 2 (0.6%) cricothyrotomy, and 14 (4.0%) tube thoracostomy. Because Japanese ELSTs are not permitted to perform such invasive procedures, none of the 667 patients transported by ELST-staffed ambulances underwent such treatment. Among patients who were transported by an EP-staffed ambulance, the two interventions and tube thoracostomy were associated with increased prehospital LOS (S2 Fig).

## Primary outcomes

Fig 2A and 2B compares preoperative time between the EP and ELST groups. The median duration of prehospital LOS was significantly longer in the EP group than in the ELST group in both crude (71.0 min vs. 40.0 min, P < 0.001) and PS-matched analyses (71.0 min vs. 41.0 min, P < 0.001). Similarly, compared with the ELST group, the median time to surgery or TAE was significantly longer in the EP group in both unmatched (187.0 min vs. 175.0 min, P < 0.001) and PS-matched analysis (189.0 min vs. 177.0 min, P = 0.002). Conversely, as shown in Fig 2C and 2D, the median ED LOS was significantly shorter in the EP group than in the ELST group in both the full cohort (119.0 min vs. 132.0 min, P < 0.001) and PS-matched cohort (120.0 min vs. 131.0 min, P = 0.043). A similar trend remained in the cumulative event rate curve analysis and log-rank tests (S3 Fig) and Cox proportional hazards regression analysis adjusted for age,

**Table 1. Demographic and clinical characteristics of injured patients requiring emergency surgery or transcatheter arterial embolization.**

| | Full cohort | | | | PS matched cohort | | | |
|---|---|---|---|---|---|---|---|---|
| | ELST (n = 667) | EP (n = 353) | P | SD (%) | ELST (n = 295) | EP (n = 295) | P | SD (%) |
| Age | | | | | | | | |
| Median (interquartile range) | 47.0 (25.0–63.0) | 56.0 (34.5–67.0) | < 0.001 | NA | 52.0 (32.0–68.0) | 54.0 (32.0–66.0) | 0.880 | NA |
| Mean ± Standard Deviation | 45.0 ± 22.9 | 51.6 ± 21.2 | < 0.001 | 29.9 | 49.9 ± 22.6 | 50.1 ± 21.3 | 0.893 | 0.9 |
| Gender | | | 0.441 | | | | 0.721 | |
| Male | 482 (72.3) | 247 (70.0) | | -5.1 | 202 (68.5) | 206 (69.8) | | 2.9 |
| Female | 185 (27.7) | 106 (30.0) | | 5.1 | 93 (31.5) | 89 (30.2) | | -2.9 |
| Admission phase | | | < 0.001 | | | | 0.757 | |
| 2002–2007 | 374 (56.1) ** | 94 (26.6) * | | -62.7 | 91 (30.8) | 90 (30.5) | | -0.7 |
| 2008–2013 | 164 (24.6) * | 159 (45.0) ** | | 44.0 | 104 (35.3) | 112 (38.0) | | 5.6 |
| 2014–2019 | 129 (19.3) * | 100 (28.3) ** | | 21.2 | 100 (33.9) | 93 (31.5) | | -5.1 |
| Trauma etiology | | | < 0.001 | | | | 0.235 | |
| Blunt | 552 (82.8) | 329 (93.2) | | 32.5 | 280 (94.9) | 273 (92.5) | | -9.8 |
| Penetrating | 115 (17.2) | 24 (6.8) | | -32.5 | 15 (5.1) | 22 (7.5) | | 9.8 |
| Anatomical severity | | | | | | | | |
| ISS | | | | | | | | |
| Median (interquartile range) | 10.0 (6.0–25.0) | 22.0 (10.0–34.0) | < 0.001 | NA | 19.0 (9.0–34.0) | 20.0 (10.0–34.0) | 0.520 | NA |
| Mean ± Standard Deviation | 16.7 ± 14.0 | 24.4 ± 15.4 | < 0.001 | 50.3 | 22.6 ± 15.9 | 22.6 ± 14.6 | 0.961 | 0 |
| AIS ($\geq$3) | | | | | | | | |
| Head or neck | 102 (15.3) | 85 (24.1) | 0.001 | 22.2 | 69 (23.4) | 60 (20.3) | 0.370 | -7.4 |
| Face | 9 (1.3) | 10 (2.8) | 0.095 | 10.4 | 6 (2.0) | 5 (1.7) | 0.761 | -2.5 |
| Chest | 166 (24.9) | 156 (44.2) | < 0.001 | 41.5 | 113 (38.3) | 119 (40.3) | 0.613 | 4.2 |
| Abdomen or pelvic contents | 131 (19.6) | 115 (32.6) | < 0.001 | 29.8 | 82 (27.8) | 89 (30.2) | 0.525 | 5.2 |
| Extremities or pelvic girdle | 343 (51.4) | 215 (60.9) | 0.004 | 19.2 | 178 (60.3) | 178 (60.3) | 1.000 | 0 |
| Physiological parameters | | | | | | | | |
| GCS score | | | < 0.001 | | | | 0.749 | |
| 13–15 | 587 (88.0) ** | 270 (76.5) * | | -30.5 | 245 (83.1) | 240 (81.4) | | -4.4 |
| 9–12 | 38 (5.7) * | 32 (9.1) ** | | 12.9 | 24 (8.1) | 22 (7.5) | | -2.5 |
| 6–8 | 26 (3.9) | 20 (5.7) | | 8.3 | 16 (5.4) | 17 (5.8) | | 1.5 |
| 4–5 | 9 (1.3) | 8 (2.3) | | 6.9 | 5 (1.7) | 7 (2.4) | | 4.8 |
| 3 | 7 (1.0) * | 23 (6.5) ** | | 29.0 | 5 (1.7) | 9 (3.1) | | 8.9 |
| SBP, mmHg | | | < 0.001 | | | | 0.622 | |
| > 89 | 586 (87.9) ** | 254 (72.0) * | | -40.5 | 236 (80.0) | 232 (78.6) | | -3.3 |
| 76–89 | 39 (5.8) * | 35 (9.9) ** | | 15.1 | 25 (8.5) | 29 (9.8) | | 4.7 |
| 50–75 | 36 (5.4) * | 47 (13.3) ** | | 27.4 | 29 (9.8) | 25 (8.5) | | -4.7 |
| 1–49 | 6 (0.9) * | 17 (4.8) ** | | 23.7 | 5 (1.7) | 9 (3.1) | | 8.9 |
| Respiratory rate, breaths/min | | | < 0.001 | | | | 0.713 | |
| > 29 | 584 (87.6) ** | 260 (73.7) * | | -35.7 | 231 (78.3) | 232 (78.6) | | 0.8 |
| 10–29 | 79 (11.8) * | 85 (24.1) ** | | 32.3 | 60 (20.3) | 61 (20.7) | | 0.8 |
| 6–9 | 4 (0.6) | 4 (1.1) | | 5.8 | 4 (1.4) | 2 (0.7) | | -6.8 |
| 1–5 | 0 (0.0) | 1 (0.3) | | 7.5 | 0 (0) | 0 (0) | | 0 |
| 0 | 0 (0.0) * | 3 (0.8) ** | | 13.1 | 0 (0) | 0 (0) | | 0 |
| RTS | | | | | | | | |
| Median (interquartile range) | 7.841 (7.550–7.841) | 7.841 (6.376–7.841) | < 0.001 | NA | 7.841 (7.108–7.841) | 7.841 (6.904–7.841) | 0.329 | NA |
| Mean ± Standard Deviation | 7.458 ± 0.899 | 6.874 ± 1.554 | < 0.001 | -46.6 | 7.241 ± 1.100 | 7.150 ± 1.195 | 0.337 | -7.8 |

*(Continued)*

**Table 1.** (Continued)

| | Full cohort | | | | PS matched cohort | | | |
|---|---|---|---|---|---|---|---|---|
| | ELST | EP | P | SD (%) | ELST | EP | P | SD (%) |
| | (n = 667) | (n = 353) | | | (n = 295) | (n = 295) | | |
| Probability of survival | | | | | | | | |
| Median (interquartile range) | 0.982 (0.950–0.996) | 0.950 (0.773–0.982) | < 0.001 | NA | 0.974 (0.837–0.995) | 0.966 (0.848–0.987) | 0.141 | NA |
| Mean ± Standard Deviation | 0.913 ± 0.191 | 0.800 ± 0.293 | < 0.001 | -44.9 | 0.853 ± 0.253 | 0.853 ± 0.239 | 0.726 | 0 |
| Charlson Comorbidity Index | | | 0.210 | | | | 0.819 | |
| 0 | 515 (77.2) | 256 (72.5) | | -10.8 | 212 (71.9) | 213 (72.2) | | 0.8 |
| 1 | 109 (16.3) | 62 (17.6) | | 3.3 | 55 (18.6) | 52 (17.6) | | -2.6 |
| 2 | 24 (3.6) | 20 (5.7) | | 9.8 | 19 (6.4) | 17 (5.8) | | -2.8 |
| ≥ 3 | 19 (2.8) | 15 (4.2) | | 7.6 | 9 (3.1) | 13 (4.4) | | 7.2 |
| Presentation time | | | 0.100 | | | | 0.520 | |
| 8:00–16:59 | 360 (54.0) * | 214 (60.6) ** | | 13.5 | 156 (52.9) | 162 (54.9) | | 4.1 |
| 17:00–23:59 | 195 (29.2) | 93 (26.3) | | -6.5 | 99 (33.6) | 87 (29.5) | | -8.8 |
| 24:00–7:59 | 112 (16.8) | 46 (13.0) | | -10.6 | 40 (13.6) | 46 (15.6) | | 5.8 |
| Presentation day | | | 0.004 | | | | 0.495 | |
| Weekdays | 459 (68.8) | 273 (77.3) | | 19.3 | 230 (78.0) | 223 (75.6) | | -5.6 |
| Weekends | 208 (31.2) | 80 (22.7) | | -19.3 | 65 (22.0) | 72 (24.4) | | 5.6 |
| Off-hours presentation[a] | 391 (58.6) | 178 (50.4) | 0.012 | -16.5 | 163 (55.3) | 161 (54.6) | 0.869 | -1.4 |
| Type of intervention | | | < 0.001 | | | | 0.808 | |
| TAE | 116 (17.4) * | 98 (27.8) ** | | 25.0 | 80 (27.1) | 76 (25.8) | | -3.1 |
| External skeletal fixation or open reduction with internal fixation | 433 (64.9) ** | 169 (47.9) * | | -34.9 | 147 (49.8) | 150 (50.8) | | 2.0 |
| Laparotomy or thoracotomy | 92 (13.8) * | 72 (20.4) ** | | 17.6 | 53 (18.0) | 58 (19.7) | | 4.3 |
| Craniotomy | 26 (3.9) | 14 (4.0) | | 0.3 | 15 (5.1) | 11 (3.7) | | -6.6 |

Data are expressed as n (%) unless otherwise noted.

** Adjusted standardized residual > 1.96.

* Adjusted standardized residual < −1.96.

[a] 17:00 to 7:59 on weekdays plus all weekend hours.

AIS, Abbreviated Injury Scale; ELST, emergency life-saving technician; EP, emergency physician; GCS, Glasgow Coma Scale; ISS, Injury Severity Score; NA, not available; PS, propensity score; RTS, Revised Trauma Score; SBP, systolic blood pressure; SD, standardized difference. TAE; transcatheter arterial embolization.

sex, ISS, GCS score, SBP, respiratory rate, and trauma etiology (S2 Table). As shown in S3 Table, injured patients who received surgery or TAE after more than 180 min (delayed intervention) were more likely to be transported by an EP-staffed ambulance and were more likely to present to the ED during off-hours. Conversely, the delayed intervention group was less likely to have shock (SBP ≤ 75 mmHg) and severe torso injury (AIS ≥ 3 injury of the abdomen or pelvic contents) and were less likely to receive TAE and laparotomy or thoracotomy.

## Other outcomes

Fig 3 and S4 Table show the differences in-hospital mortality between EP-staffed ambulances and ELST-staffed ambulances. In PS-matched patients, there were no significant differences for in-hospital mortality between the EP and ELST groups (8.8% vs.9.8%; odds ratio [OR] 0.89; 95% CI 0.51–1.55). A similar trend was observed with other statistical assumptions, such as in the logistic regression model using PS as an explanatory variable (adjusted OR 0.73; 95% CI 0.43–1.26) and IPTW analysis (OR 0.77; 95% CI 0.47–1.26).

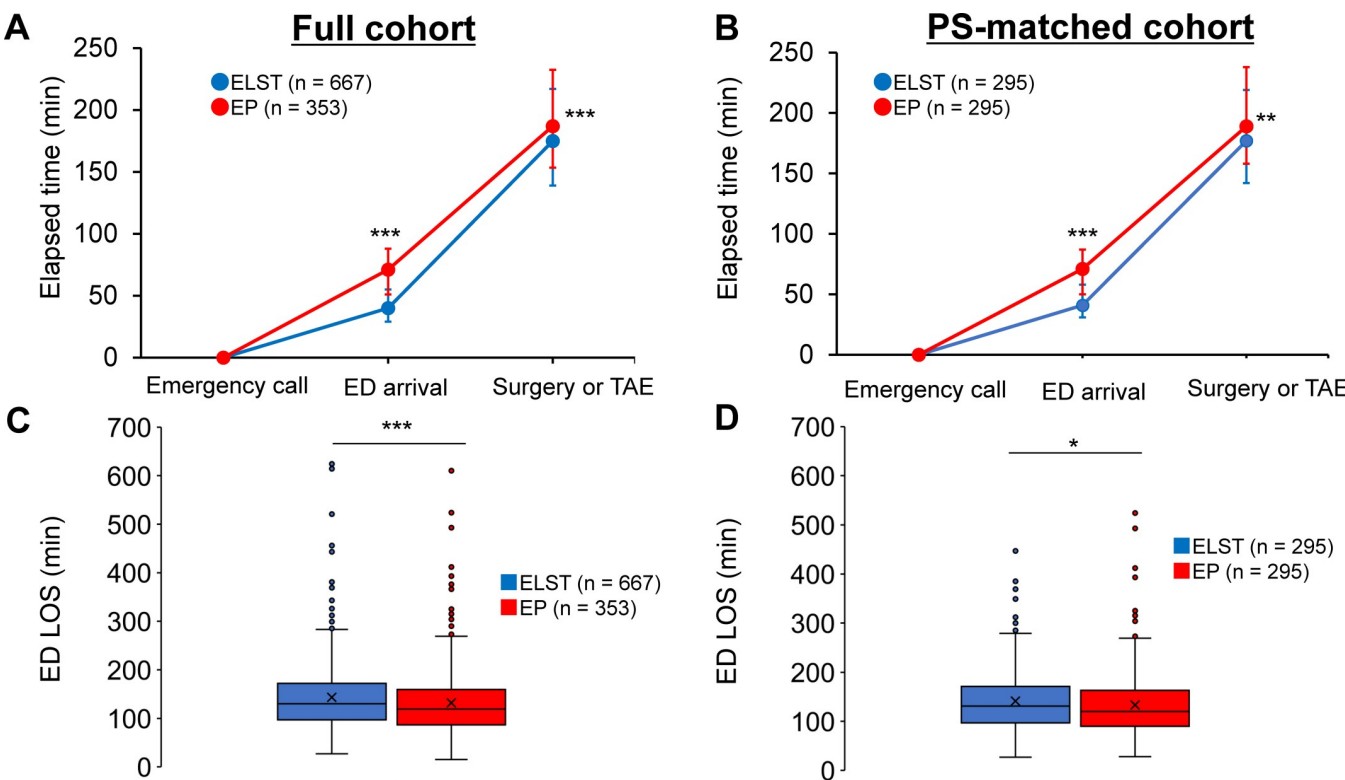

**Fig 2. Preoperative time course in injured patients requiring emergency surgery or TAE: EP-staffed vs. ELST-staffed ambulance.** Elapsed time from emergency call to ED arrival and arrival in the operating room or catheterization laboratory in the full cohort (**A**) and PS-matched cohort (**B**). Circles represent the median, and error bars represent the 25th and 75th percentiles. Box plots of ED LOS in the full (**C**) and PS-matched (**D**) cohorts. ED LOS is defined as time from ED arrival to operating room or catheterization laboratory arrival. Solid line inside the box represents the median, x represents the mean, the box represents the 25th and 75th percentiles, the whiskers represent the lower and upper extremes, and the circles represent outliers. ***P < 0.001, **P < 0.01, and *P < 0.05 using the Mann–Whitney U test. ED, emergency department; ELST, emergency life-saving technician; EP, emergency physician; LOS, length of stay; TAE, transcatheter arterial embolization.

By contrast, among patients who underwent laparotomy or thoracotomy, the EP group had a decreased odds for mortality as compared with the ELST group (adjusted OR 0.29; 95% CI 0.10–0.85). In the remaining subgroups, there were no significant differences for in-hospital mortality between two groups (S4 Fig).

The S5 Fig shows the distribution of time to death. Nearly two-thirds of deaths occurred within 24 hours (early deaths, the second peak in the trimodal distribution [34–37]), and the time distributions did not differ between the EP group and ELST group.

When considering hospital LOS and patient disposition among survivors, the median hospital LOS was significantly longer in the EP group than in the ELST group in both the full cohort (36.0 days vs. 27.0 days, P = 0.0019; Fig 4A) and PS-matched cohort (38.0 days vs. 29.0 days, P = 0.0073; Fig 4B). As shown in Fig 4C, fewer patients in the EP group were discharged home (61.6%) than in the ELST group (74.5%) in the full cohort. In the PS-matched cohort (Fig 4D), patient dispositions were similar between the two groups.

## Discussion

In this single-site observational study, we found that EP involvement in prehospital trauma care was associated with an increased prehospital LOS and delayed definitive hemostasis among injured patients. These associations were consistent with both the crude and adjusted

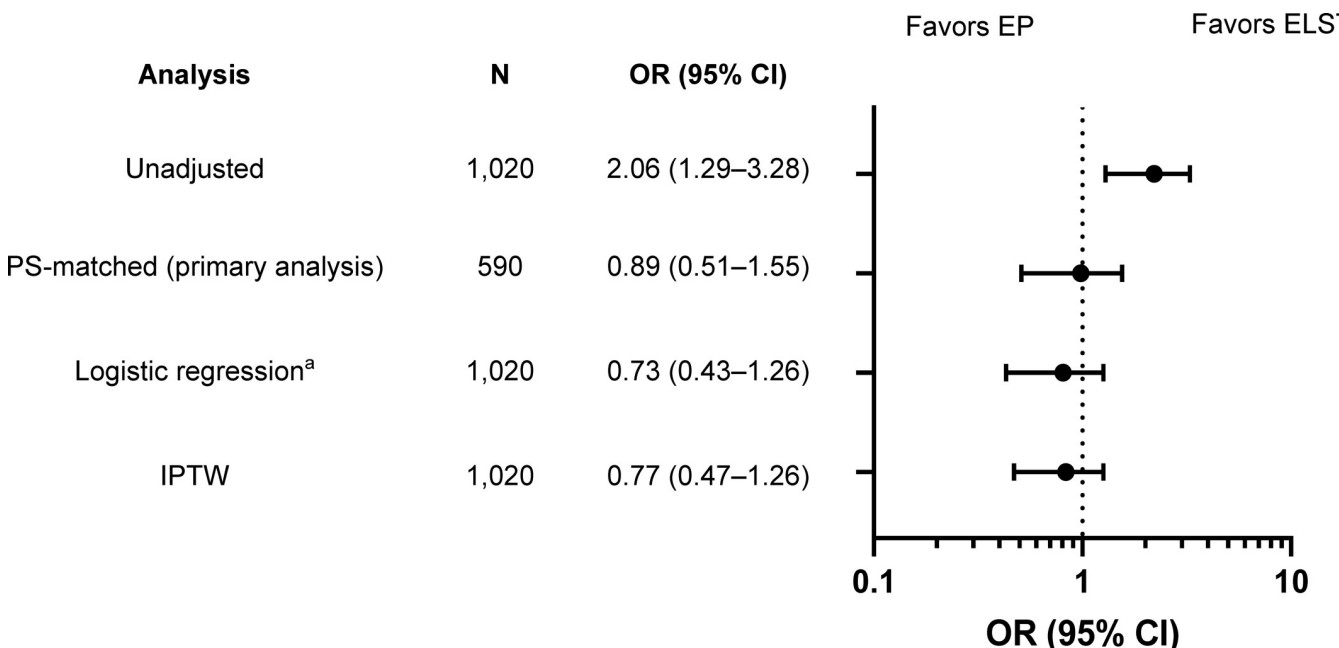

**Fig 3. Odds ratios for in-hospital mortality among injured patients: EP-staffed ambulance versus ELST-staffed ambulance.** The reference set was the group of injured patients transported to the ED by ELST-staffed ambulances. [a]Adjusted for PS as described in the Methods. CI, confidence interval, ELST, emergency life-saving technician; EP, emergency physician; IPTW, inverse probability of treatment weighting; OR, odds ratio; PS, propensity score.

analyses. In contrast, there was a slightly but significant reduction in ED LOS among injured patients who were transported by EP-staffed ambulances. No significant survival differences were observed between the two groups. EP involvement in prehospital trauma care was also associated with increased hospital LOS and did not alter disposition locations.

Increased prehospital LOS and total time to surgery or TAE in the EP group can likely be explained by the increased number of prehospital interventions. Alternatively, in long-distance transfers, the regional medical control center or ELST at the scene may tend to request dispatch of an EP-staffed ambulance to initiate early advanced trauma care. Prehospital LOS and the number of prehospital advanced procedures in this study were comparable to other Japanese EP-staffed emergency medical services [14, 20] and those abroad [5, 8, 13, 18, 21]. As with most EP-staffed medical services, prehospital hemostatic procedures for major bleeding would not be practical at this study site owing to a lack of resources. Our results showed that deaths predominantly occurred within the first day in both the ELST group and EP group, corresponding to the second peak (deaths within 24 hours after trauma) of the classic trimodal trauma–death distribution [34–37]. The main cause of the second peak in mortality among injured patients needing definitive intervention is known to be hemorrhagic shock [34–37]. Similar time-of-death distributions between the EP and ELST groups suggest that EP involvement in prehospital trauma care does not prevent death from hemorrhagic shock. A number of prehospital interventions are known to be positively correlated with prolonged prehospital LOS, that is, these are reported to be related to a greater risk of poor outcomes [5, 38]. Thus, EPs dispatched to the scene must consider the trade-off between safely performing crucial interventions and limiting time on the scene, particularly in injured patients who require definitive hemostasis.

Turning to survival outcome, in this cohort study, we failed to show a difference in mortality between EP-staffed ambulances and ELST ambulances. These results were consistent in

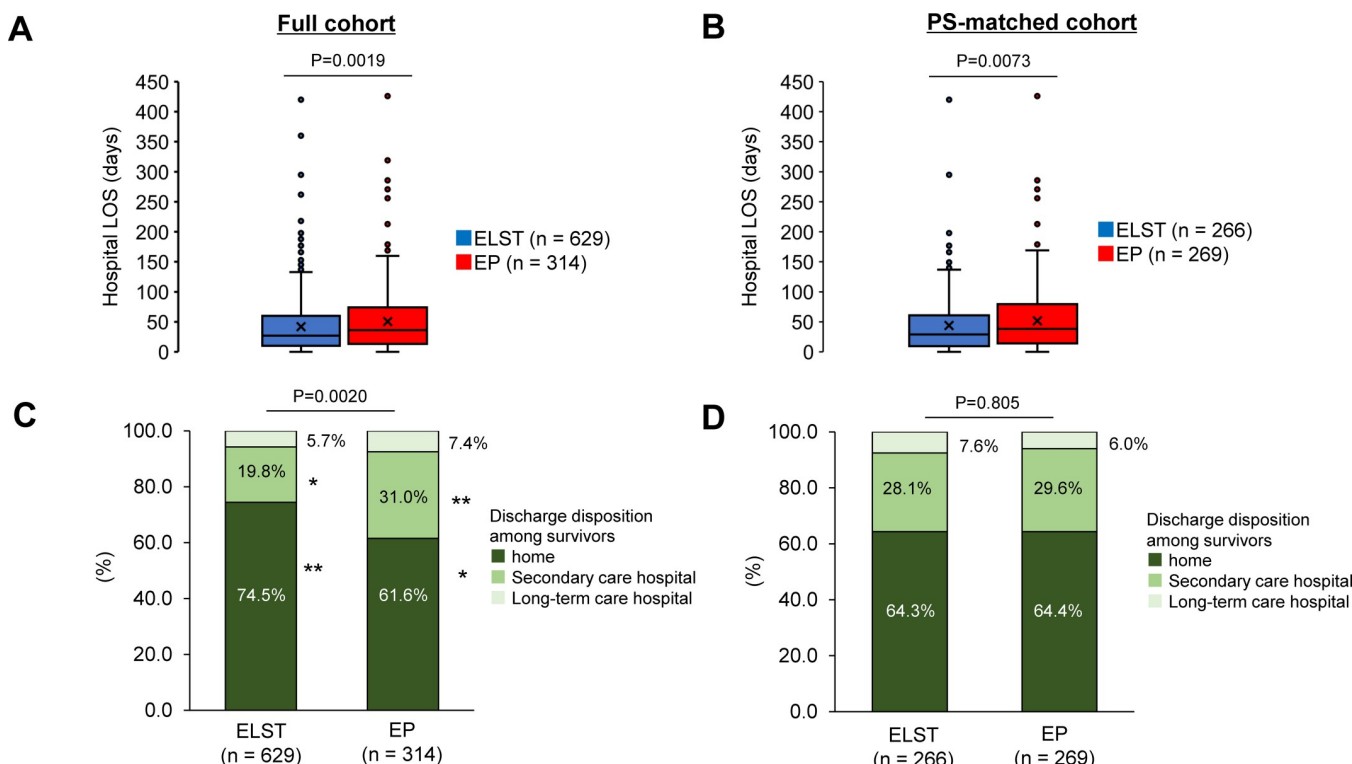

**Fig 4. Differences in hospital LOS and disposition location among survivors: EP-staffed ambulance versus ELST-staffed ambulance.** Hospital LOS among survivors in the full (**A**) and PS-matched (**B**) cohorts. Solid line inside the box represents the median, x represents the mean, the box represents the 25th and 75th percentiles, the whiskers represent the lower and upper extremes, and the circles represent outliers. Disposition location of survivors in the full (**C**) and PS-matched (**D**) cohorts. **Adjusted standardized residual > 1.96 and *adjusted standardized residual < −1.96. ELST, emergency life-saving technician; EP, emergency physician; LOS, length of stay; PS, propensity score.

subanalysis using the IPTW method and a logistic regression model adjusted for PS. In the PS-matched cohort, the discharge disposition of survivors was also similar between the two groups, suggesting that EP involvement in prehospital trauma care did not affect the functional outcome of patients with trauma. Delays in hospital arrival and definitive treatment is one plausible explanation for the failure of having an EP at the prehospital scene to improve survival and functional outcomes. These results were in agreement with recent studies including among hypotensive patients with trauma who have SBP < 90 mmHg [24] and severely injured patients with ISS > 15 [20] who were transported by EP-staffed ambulances. These results, together with the current findings, collectively suggest there are no major clinical benefits with EP involvement in prehospital care regarding survival, as compared with ELSTs, especially for injured patients requiring definitive hemostatic treatment.

In this study, we also found that after PS-matching, hospital LOS was somewhat increased in the EP group as compared with the ELST group among trauma survivors. Any delay in the definitive control of hemorrhage can result in hypovolemic shock, coagulopathy, multiple organ failure, immunosuppression, and secondary infection [39–42], all of which can result in prolonged hospital stay. Because participants in this study were severely injured and required surgical hemostasis, unfavorable effects of delayed hemostasis owing to prolonged prehospital care provided by EP-staffed ambulances might be present.

We also observed that there was a slight, but significant, reduction in the time from ED arrival to arrival in the operating room or catheterization laboratory when an EP was

dispatched to the prehospital scene. At this study site, if an injured patient is expected to require emergency surgery or TAE, the on-scene EP provides their assessment to the ED using a dedicated direct line. Specialists such as trauma surgeons, anesthesiologists, and interventional radiologists are usually present in the ED before the patient's arrival. Such accurate diagnosis provided by on-scene EP and cooperation between prehospital and in-hospital trauma teams is the one plausible explanation of the observed findings. Similarly, in a study of other Japanese EP-staffed emergency medical services, the time to blood transfusion was significantly shorter in patients in the EP group than in the ELST group [20]. Accurate prehospital information and earlier clinical decision making regarding injured patients provided by an on-scene EP [43] would explain the above findings.

Subgroup analyses in this study revealed that injured patients requiring laparotomy or thoracotomy benefited from an EP-staffed ambulance in terms of survival. If patients have extremity and/or pelvic injuries, on-scene ELSTs attempt to control bleeding by applying manual compression, a tourniquet, or a pelvic sling belt; however, these measures are not attempted by ELSTs in patients with intraabdominal or intrathoracic injury. EP-specific prehospital interventions, such as tube thoracostomy, ultrasound examination, and accurate clinical judgement might be beneficial for injured patients with non-compressible sources of bleeding [44]. Future larger studies will be needed to further clarify the usefulness of EP-staffed ambulances in this patient population.

## Limitations and strengths

Several limitations of the current study should be acknowledged. Firstly, this study was performed at a single site, limiting the generalizability of the findings. Secondly, although rigorous adjustments were made in a PS-matched analysis, other unmeasured factors may have confounded our results, as with any observational study. For example, our survey did not collect information such as distance to the scene or patient body mass index [45], insurance status, or use of vasopressors [46]. Further analyses including such variables will be required to further clarify the association between EP involvement in prehospital trauma care and measured clinical outcomes. Thirdly, our database did not contain information on some prehospital advanced procedures such as ultrasound examination and fluid resuscitation, which might independently affect the outcome. For example, there would be a difference in prehospital intravenous fluid administration between EP and ELST groups, which might affect the outcomes of patients with trauma [47–49]. Finally, we did not calculate the sample size in advance. As described in the Methods, the retrospective nature of the study predetermined the sample size. However, a post-hoc power calculation demonstrated that the power of our study was sufficient (power > 0.80) for all primary outcomes examined.

Despite these limitations, the current study also had several strengths. Firstly, this study captured in-depth information such as presentation time, presentation day, and type of definitive intervention. Indeed, these variables were important confounders not measured in previous studies [13–24]. Additionally, this study clarified the difference in hospital LOS and discharge disposition between EP-staffed ambulances and ELST-staffed ambulances. Prior to our study, little was known regarding whether and how EP dispatch to the prehospital scene affects such relevant trauma care metrics [13–24]. Secondly, the data for this study were gathered from existing electronic databases, and all data were entered into the database by one trained emergency physician only (author K.S.). The measured outcomes were objective (i.e., prehospital, ED and hospital LOS, location of patient dispositions) and less prone to diagnostic errors, minimizing the risk of ascertainment bias. Additionally, there were no missing data for all relevant analyses. Most previous investigations comparing the outcomes of injured patients transported by EP-

staffed and ELST-staffed ambulances used large trauma databases, but a considerable amount of data was missing in those studies [15, 17, 20, 24]. To handle missing data, previous investigations have eliminated all missing data [15, 20, 24] or used multiple imputation [17], but both methods can generate bias. Furthermore, to mitigate the risk of biased assessment, the author who constructed the database (K.S.) was not involved in the any of the statistical analysis.

## Conclusions

At a community hospital in Japan, EP-staffed ambulances were associated with prolonged prehospital stay and delay in definitive treatment among injured patients who were in need of emergency surgery or TAE, in comparison with ELST-staffed ambulances. EP-staffed ambulances also did not improve survival outcomes in this trauma subset. Among survivors, EP involvement in prehospital trauma care was associated with increased hospital LOS and did not alter disposition locations. Our observations provide an opportunity to reconsider the use of EP-staffed ambulances in this population.

## Supporting information

**S1 Data. Minimal anonymized data set used in this study.**
(XLSX)

**S1 Table. Comparison of prehospital intervention: EP-staffed ambulance versus ELST-staffed ambulance.** Data are expressed as n (%). ELST, emergency life-saving technician; EP, emergency physician; PS, propensity score.
(PDF)

**S2 Table. Cox proportional hazards regression analysis of cumulative preoperative time: EP-staffed ambulance versus ELST-staffed ambulance.** Cox proportional hazards regression analyses were performed with adjustment for age, sex, Injury Severity Score, Glasgow Coma Scale score, systolic blood pressure, respiratory rate, and trauma etiology. The reference set was the group of injured patients transported to the ED by ELST-staffed ambulances. CI, confidence interval; ED, emergency department; ELST, emergency life-saving technician; EP, emergency physician; TAE, trans arterial embolization.
(PDF)

**S3 Table. Factors associated with delayed definitive intervention[a].** Data are expressed as n (%) or median (interquartile range). **Adjusted standardized residual > 1.96. *Adjusted standardized residual < −1.96. [a]Time from the emergency call to arrival in the operating room or catheterization laboratory later than 180 min. [b]17:00 to 7:59 on weekdays plus all weekend hours. AIS, Abbreviated Injury Scale; ELST, emergency life-saving technician; EP, emergency physician; GCS, Glasgow Coma Scale; ISS, Injury Severity Score; NA, not available; PS, propensity score; RTS, Revised Trauma Score; SBP, systolic blood pressure; SD, standardized difference. TAE; transcatheter arterial embolization.
(PDF)

**S4 Table. Comparison of mortality rate: EP-staffed ambulance versus ELST-staffed ambulance.** Data are expressed as n (%). ELST, emergency life-saving technician; EP, emergency physician; PS, propensity score.
(PDF)

**S1 Fig.** Distribution of PS in the full (A) and PS-matched (B) cohorts. ELST, emergency life-saving technician; EP, emergency physician; PS, propensity score.
(PDF)

**S2 Fig. Prehospital LOS according to the number and type of interventions in EP-staffed ambulances.** Column scatter plots representing the data distribution (circles), median (horizontal bar), and interquartile range (vertical bar). **P < 0.01, *P < 0.05. The P values were derived using the Kruskal–Wallis test followed by Dunn's post hoc tests with Bonferroni correction. EP: emergency physician; LOS: length of stay.
(PDF)

**S3 Fig. Comparison of cumulative incidence rate curves: EP-staffed ambulance versus ELST-staffed ambulance.** Cumulative incidence rate curves from emergency call to ED arrival in the full **(A)** and PS-matched **(B)** cohorts. Cumulative incidence rate curves from ED arrival to arrival in the operating room or catheterization laboratory in the full **(C)** and PS-matched **(D)** cohorts. Cumulative incidence rate curves from emergency call to surgery or TAE in the full **(E)** and PS-matched **(F)** cohorts. ***P < 0.001, **P < 0.01, *P < 0.05 by log-rank test. ED, emergency department; ELST, emergency life-saving technician; EP, emergency physician; *PS*, propensity score; TAE, transarterial embolization.
(PDF)

**S4 Fig. Subgroup analysis of hospital mortality in injured patients requiring emergency surgery or TAE: EP-staffed ambulance versus ELST-staffed ambulance.** The reference set was the group of injured patients transported to the ED by ELST-staffed ambulances. [a]Adjusted for PS, as described in the Methods. [b]17:00 to 7:59 on weekdays plus all weekend hours. CI, confidence interval; ELST, emergency life-saving technician; EP, emergency physician; GCS, Glasgow Coma Scale; ISS, Injury Severity Score; OR, odds ratio; SBP, systolic blood pressure; TAE, transcatheter arterial embolization.
(PDF)

**S5 Fig. Time-of-death analysis: EP-staffed ambulance versus ELST-staffed ambulance.** The distribution of time to death in full cohort **(A)** and PS-matched **(B)** cohort. The proportion of deaths in the full cohort **(C)** and PS-matched cohort **(D)** within 24 hours. ELST, emergency life-saving technician; EP, emergency physician; LOS, length of stay; PS, propensity score.
(PDF)

## Acknowledgments

We thank our colleagues at Ohta Nishinouchi Hospital for the data acquisition. We also thank Nozomi Ono, MD (Department of Psychiatry, Hoshigaoka Hospital, Koriyama, Japan) for providing assistance in reviewing the manuscript. Finally, we thank Analisa Avila, MPH, ELS, from Edanz (https://jp.edanz.com/ac), for editing a draft of this manuscript.

## Author Contributions

**Conceptualization:** Yuko Ono, Kazuaki Shinohara.

**Data curation:** Yuko Ono, Yudai Iwasaki, Takaki Hirano, Katsuhiko Hashimoto, Takeyasu Kakamu.

**Formal analysis:** Yuko Ono, Takeyasu Kakamu.

**Investigation:** Yuko Ono, Yudai Iwasaki, Takaki Hirano, Katsuhiko Hashimoto.

**Methodology:** Yuko Ono, Takeyasu Kakamu.

**Project administration:** Yuko Ono, Shigeaki Inoue, Joji Kotani, Kazuaki Shinohara.

**Supervision:** Shigeaki Inoue, Joji Kotani, Kazuaki Shinohara.

**Validation:** Takeyasu Kakamu.

**Visualization:** Yuko Ono.

**Writing – original draft:** Yuko Ono.

**Writing – review & editing:** Yudai Iwasaki, Takaki Hirano, Katsuhiko Hashimoto, Takeyasu Kakamu, Shigeaki Inoue, Joji Kotani, Kazuaki Shinohara.

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
