## [Decision Letter · Decision Letter 0]

16 Aug 2021

PONE-D-21-20967

Impact of emergency physician-staffed ambulances on preoperative time course and survival among injured patients requiring emergency surgery or transarterial embolization: A retrospective cohort study at a community emergency department in Japan

PLOS ONE

Dear Dr. Ono,

Thank you for submitting your manuscript to PLOS ONE. After careful consideration, we feel that it has merit but does not fully meet PLOS ONE’s publication criteria as it currently stands. Therefore, we invite you to submit a revised version of the manuscript that addresses the points raised during the review process.

We look for your timely response and corresponding revisions based on our reviewers' comments.

We look forward to receiving your revised manuscript.

Kind regards,

Chun Chieh Yeh, M.D., Ph.D.

Academic Editor

PLOS ONE

Journal Requirements:

2. We note that you have indicated that data from this study are available upon request. PLOS only allows data to be available upon request if there are legal or ethical restrictions on sharing data publicly.

For more information on unacceptable data access restrictions, please see http://journals.plos.org/plosone/s/data-availability#loc-unacceptable-data-access-restrictions.

Additional Editor Comments:

Thanks for your submission for this interesting article. Our reviewers gave some comments and we look for your timely and appropriate response and revisions.

Reviewers' comments:

Reviewer's Responses to Questions

**Comments to the Author**

1. Is the manuscript technically sound, and do the data support the conclusions?

Reviewer #1: Yes

Reviewer #2: Yes

2. Has the statistical analysis been performed appropriately and rigorously? 

Reviewer #1: Yes

Reviewer #2: Yes

3. Have the authors made all data underlying the findings in their manuscript fully available?

Reviewer #1: Yes

Reviewer #2: Yes

4. Is the manuscript presented in an intelligible fashion and written in standard English?

Reviewer #1: Yes

Reviewer #2: Yes

5. Review Comments to the Author

Reviewer #1: I congratulate the authors on performing this study and thank the editor of PLOS ONE for offering me the opportunity to review this interesting study.

This propensity matched analysis focusses on the physician vs paramedic staffed prehospital ambulance service for severely injured population. Historically in trauma patients early treatment ideally in the so called Golden hour is regarded key to improve the outcome of patients. Therefore the preoperative time to definitive intervention of almost 3 hours in both groups seems rather long. Could the authors be looking at a subset of patients who have been able to survive such a long interval from the time of injury until definitve treatment?

Especially since the special interventions that are performed by the EP staffed ambulance can be mainly regarded to stabilize A and B problems (ATLS; Airway, Breathing) rather then haemmorrhage as a C-problem. For patients at risk of exsanguination (prehospital) care should be focused towards stopping the bleeding and if this cannot be achieved a scoop and run to the nearest trauma centre is the main priority.

To my knowledge the application of resuscitative endovascular balloon occlusion of the aorta (REBOA) is widely practiced by emergency physicians on the ED in Japan. Do the authors see a role of prehospital application of such an advanced hemostatic intervention?

Throughout the text you mention prehospital interventions performed bij EP staffed teams; on of these include "tube thoracotomy". I assume you mean tube thoracoStomy and not a resuscitative thoracotomy to gain acces to the thorax to perform supradiafragmatic aortic crossclaming or other major surgical interventions given the context of the sentences. Please change; line 71, line 122, line 133, line 271, 365.

Perioperative time should be corrected to PRE operative time since it does not include the time of the definitive surgical intervention and/or post postprocedural time.

Please reduce the length of the manuscript which is too long and avoid iterations.

For example at the end of the introduction paragraph the authors mentions the primary study outcome and hypothesis AND also secondary outcomes. I suggest to remove line 89 we also compared ..... that supports our hypothesis. line 93. In addition this last sentence is a result/conclusion and does not belong in the introduction paragraph.

Please avoid to give the results both described in text and in the tables. Stick to the primary outcomes in text and refer to the tables.

The first paragraph of the discussion is a repitition of the rationale/method. Please begin with stating the most significant finding which in my opinion starts at line 353 Both crude and adjusted... among injured patients.

line 359-360 repitition

The methods and discussion paragraph are quite long. The dicussion is 6.5 pages, i would suggest to reduce the length by 50% to increase the readibility. Also recommend to shorten the methods section.

Reviewer #2: Thank you for giving me the opportunity to review the manuscript. This study evaluated the effects of physician staffed ambulance on outcomes (time course, mortality, and etcetera). As recent observational study demonstrated, this study result and conclusion was consistent with it that EP-staffed ambulances were found to be associated

with prolonged prehospital time, delay in definitive treatment. And I as a reviewer agreed with this notion. The manuscript was well constructed and was easily to understand. The one limitation of this study was a single centered observational study, and this point was already noted. I recommend the authors to add minor information. What prehospital treatment was associated with prolonged prehospital stay? And the authors could evaluate the time course between prehospital intervention+ and prehospital intervention - and EP? These point could not solved from the multicentered observational study such as Ref 20 and 24. And this point may add some novelty.

6. PLOS authors have the option to publish the peer review history of their article (what does this mean?). If published, this will include your full peer review and any attached files.

Reviewer #1: No

Reviewer #2: **Yes: **Makoto Aoki

---

## [Author Response · Author response to Decision Letter 0]

7 Sep 2021

Dr. Chun Chieh Yeh 

Academic Editor 

PLOS ONE

September 7, 2021

Re: Revised version of manuscript ID # PONE-D-21-20967

Dear Dr. Yeh,

My coauthors and I would like to resubmit our manuscript entitled “Impact of emergency physician-staffed ambulances on preoperative time course and survival among injured patients requiring emergency surgery or transarterial embolization: A retrospective cohort study at a community emergency department in Japan,” to be considered for publication as an original research article in PLOS ONE. We are grateful to the reviewers for their valuable comments and suggestions, which we have incorporated into the revised (R1) version of the manuscript. Their thorough review has greatly increased the scientific value and readability of our manuscript. 

Our point-by-point responses to the reviewers’ comments are provided below. Please note that any changes made in the R1 version appear in red. We hope that, with these amendments, our revised manuscript is suitable for publication in PLOS ONE.

We thank you again for your time and effort in considering our revised manuscript and look forward to hearing from you at your earliest convenience.

Yours sincerely,

Yuko Ono, MD, PhD

Department of Disaster and Emergency Medicine, Graduate School of Medicine

Kobe University, Kobe city, Hyogo 650-0017, Japan 

Tel: +81-078-382-6521, Fax: +81-078-341-5254 

Email: windmill@people.kobe-u.ac.jp

Point-by-point responses to the reviewers’ comments

Additional Editor 

Response:

We have ensured that our revised manuscript meets PLOS ONE's style requirements, as provided above.

Response:

Thank you for drawing this problem to our attention. As requested, the minimal anonymized data set is now included in the R1 manuscript (S1 Data in the supporting information file).

Additional Editor Comments:

Thanks for your submission for this interesting article. Our reviewers gave some comments and we look for your timely and appropriate response and revisions.

Response:

We greatly appreciate your time and consideration. We have thoroughly revised our previous manuscript based on both reviewers’ suggestions. We hope that our revisions now fulfil your expectations and you find our manuscript suitable for publication in PLOS ONE.

 

To Reviewer #1: 

We greatly appreciate your time and effort in reviewing our manuscript and providing your insightful comments. We have incorporated the recommended changes into the revised (R1) version of our manuscript. Any changes based on your comments are indicated in red.

I congratulate the authors on performing this study and thank the editor of PLOS ONE for offering me the opportunity to review this interesting study. This propensity matched analysis focusses on the physician vs paramedic staffed prehospital ambulance service for severely injured population. Historically in trauma patients early treatment ideally in the so called Golden hour is regarded key to improve the outcome of patients. Therefore the preoperative time to definitive intervention of almost 3 hours in both groups seems rather long. Could the authors be looking at a subset of patients who have been able to survive such a long interval from the time of injury until definitive treatment?

Response:

We agree with the reviewer's advice and have performed additional analysis. Accordingly, we have generated a new table (S3 Table in the supporting information file) and amended the Methods (page 13, lines 207-210) and Results (page 24, lines 277-282). 

To determine factors associated with delayed definitive intervention (time from emergency call to surgery or transarterial embolization [TAE] > 180 min), differences in the characteristics between patients who received surgery or TAE within 180 min and those who received these interventions later were compared. We found that injured patients who received surgery or TAE later than 180 min were more likely to be transported by an emergency physician (EP)-staffed ambulance and more likely to present to the emergency department (ED) during off-hours. Conversely, this patient population was less likely to have shock (systolic blood pressure [SBP] ≤ 75 mmHg) and severe torso injury (Abbreviated Injury Scale (AIS) ≥ 3 injury of the abdomen or pelvic contents) and less likely to receive TAE and laparotomy or thoracotomy (S3 Table in the supporting information file). There are several plausible explanations for the observed findings. Blood from massive bleeding is likely to collect in the thoracic, pelvic, and abdominal cavities, and patients with severe torso injury are likely to have shock. All trauma care staff should work together on such high-risk patients in the ED, bringing them immediately to the operating room or catheterization laboratory according to their risks. Like most Japanese tertiary medical centers, surgeons, anesthesiologists, and interventional radiologists in the study hospital are more likely to respond from outside the hospital for emergency trauma surgery or TAE during off-hours. Additionally, examination and resuscitative treatment of severely injured patients require many staff, and decreases in staffing during off-hours may become more apparent. 

As the reviewer also pointed out, the concept that definitive trauma care must be initiated within a 60-minute window (the golden hour) is pervasive [1, 2]. However, as past studies have indicated, providing care during the golden hour is often impossible [3, 4], especially in rural areas like the one in which the study hospital is located [5, 6]. The prehospital length of stay (LOS) or time to definitive treatment in this study were comparable to those of other facilities in Japan [7, 8] and in other countries [9-13].

Especially since the special interventions that are performed by the EP staffed ambulance can be mainly regarded to stabilize A and B problems (ATLS; Airway, Breathing) rather then hemorrhage as a C-problem. For patients at risk of exsanguination (prehospital) care should be focused towards stopping the bleeding and if this cannot be achieved a scoop and run to the nearest trauma centre is the main priority. To my knowledge the application of resuscitative endovascular balloon occlusion of the aorta (REBOA) is widely practiced by emergency physicians on the ED in Japan. Do the authors see a role of prehospital application of such an advanced hemostatic intervention?

Response:

The reviewer noted a very important point. As indicated, the usefulness of resuscitative endovascular balloon occlusion of the aorta (REBOA) in severely injured patients has been proposed recently [14-17]. At the study site, however, REBOA is usually performed in the ED or catheterization laboratory under radiographic guidance, and no patients received this intervention at the prehospital scene (R1 manuscript, page 9 lines 127-130, and S1 Table in the supporting information file). 

In the ED or catheterization laboratory, 27 of 1020 (2.6%) study participants received REBOA during the study period. However, it is not possible to evaluate the usefulness of REBOA based on this small sample size.

Throughout the text you mention prehospital interventions performed by EP staffed teams; on of these include "tube thoracotomy". I assume you mean tube thoracoStomy and not a resuscitative thoracotomy to gain access to the thorax to perform supradiafragmatic aortic crossclaming or other major surgical interventions given the context of the sentences. Please change; line 71, line 122, line 133, line 271, 365.

Perioperative time should be corrected to PRE operative time since it does not include the time of the definitive surgical intervention and/or post postprocedural time.

Response:

The reviewer is right to point out that the phrases "tube thoracotomy" and "perioperative time" are inappropriate. We therefore replaced all instances of "tube thoracotomy" with "tube thoracostomy" and "perioperative time" to "preoperative time" throughout the revised manuscript. 

Please reduce the length of the manuscript which is too long and avoid iterations.

For example, at the end of the introduction paragraph the authors mention the primary study outcome and hypothesis AND also secondary outcomes. I suggest to remove line 89 we also compared ..... that supports our hypothesis. line 93. In addition, this last sentence is a result/conclusion and does not belong in the introduction paragraph.

Please avoid to give the results both described in text and in the tables. Stick to the primary outcomes in text and refer to the tables.

The first paragraph of the discussion is a repetition of the rationale/method. Please begin with stating the most significant finding which in my opinion starts at line 353 Both crude and adjusted... among injured patients.

line 359-360 repetition

The methods and discussion paragraph are quite long. The discussion is 6.5 pages, i would suggest to reduce the length by 50% to increase the readability. Also recommend to shorten the methods section.

Response:

We agree with the reviewer's advice and have removed the redundant parts from the R1 manuscript. The deleted portions are as follows:

Original version, page 6, line 87 to page 7, line 93; page 9, lines 134 to 138; page 12, lines 182 to 184 and lines 190 to 191; page 15, lines 245 to 247; page 24, lines 282 to 283; page 25, lines 287 to 291; page 27; lines 331 to 333; page 28; line 348 to 353; page 29, lines 359 to 362.

Accordingly, we have successfully reduced the manuscript by approximately 500 words (2-page length). We greatly appreciate your comments, which have increased the readability of our manuscript. 

To Reviewer #2: 

Thank you very much for taking the time to review our manuscript and for your pertinent suggestions. We have followed your advice and generated a new figure (S2 Fig in the supporting information file) and the accompanying discussion. We have incorporated your recommended changes into the R1 version of the manuscript, which appear in red font.

Reviewer #2: Thank you for giving me the opportunity to review the manuscript. This study evaluated the effects of physician staffed ambulance on outcomes (time course, mortality, and etcetera). As recent observational study demonstrated, this study result and conclusion was consistent with it that EP-staffed ambulances were found to be associated with prolonged prehospital time, delay in definitive treatment. And I as a reviewer agreed with this notion. The manuscript was well constructed and was easily to understand. The one limitation of this study was a single centered observational study, and this point was already noted.

Response:

We thank the reviewer very much for positively evaluating our manuscript.

I recommend the authors to add minor information. What prehospital treatment was associated with prolonged prehospital stay? And the authors could evaluate the time course between prehospital intervention+ and prehospital intervention - and EP? These points could not be solved from the multicentered observational study such as Ref 20 and 24. And this point may add some novelty.

Response:

We greatly appreciate this pertinent advice. To address this comment, we examined the differences in prehospital LOS according to the number and type of interventions in EP-staffed ambulances. We found that the two interventions and tube thoracostomy were associated with increased prehospital LOS (S2 Fig in the supporting information file, and R1 manuscript page 23, lines 261-263). Because the number of prehospital interventions and prolonged prehospital LOS are both reported to be associated with a greater risk of poor outcomes [9, 18–21], these new results will be useful for health care professionals who are involved in prehospital trauma care. EPs dispatched to the scene must therefore consider the trade-off between safely performing crucial interventions and limiting the time on the scene (R1 manuscript, page 28, lines 359-361). 

Other amendments

We added several pertinent references (R1 manuscript, Refs 45 and 46) to support our claims that appear in the discussion (page 31, lines 408-409).

References

1. Lerner EB, Moscati RM. The golden hour: scientific fact or medical “urban legend”? Acad Emerg Med 2001;8:758–60.

2. Newgard CD, Schmicker RH, Hedges JR, Trickett JP, Davis DP, Bulger EM, et al. Emergency medical services intervals and survival in trauma: assessment of the "golden hour" in a North American prospective cohort. Ann Emerg Med. 2010;55:235–46.e4. 

3. Wyen H, Lefering R, Maegele M, Brockamp T, Wafaisade A, Wutzler S, et al. The golden hour of shock - how time is running out: prehospital time intervals in Germany--a multivariate analysis of 15, 103 patients from the TraumaRegister DGU(R). Emerg Med J. 2013;30:1048–55. 

4. Al-Thani H, Mekkodathil A, Hertelendy AJ, Frazier T, Ciottone GR, El-Menyar A. Prehospital Intervals and In-Hospital Trauma Mortality: A Retrospective Study from a Level I Trauma Center. Prehosp Disaster Med. 2020;35:508–15.

5. Ono Y, Yokoyama H, Matsumoto A, Kumada Y, Shinohara K, Tase C. Is preoperative period associated with severity and unexpected death of injured patients needing emergency trauma surgery? J Anesth. 2014;28:381–9.

6. Gonzalez RP, Cummings GR, Phelan HA, Mulekar MS, Rodning CB. Does increased emergency medical services prehospital time affect patient mortality in rural motor vehicle crashes? A statewide analysis. Am J Surg. 2009;197:30–4. 

7. Hirano Y, Abe T, Tanaka H. Efficacy of the presence of an emergency physician in prehospital major trauma care: A nationwide cohort study in Japan. Am J Emerg Med. 2019;37:1605–10.

8. Tsuchiya A, Tsutsumi Y, Yasunaga H. Outcomes after helicopter versus ground emergency medical services for major trauma?propensity score and instrumental variable analyses: a retrospective nationwide cohort study. Scand J Trauma Resusc Emerg Med. 2016;24:140. 

9. Gauss T, Ageron FX, Devaud ML, Debaty G, Travers S, Garrigue D, et al. Association of Prehospital Time to In-Hospital Trauma Mortality in a Physician-Staffed Emergency Medicine System. JAMA Surg. 2019;154:1117–24. 

10. Brown JB, Rosengart MR, Forsythe RM, Reynolds BR, Gestring ML, Hallinan WM, et al. Not all prehospital time is equal: Influence of scene time on mortality. J Trauma Acute Care Surg. 2016;81:93–100. 

11. Maddock A, Corfield AR, Donald MJ, Lyon RM, Sinclair N, Fitzpatrick D, et al. Prehospital critical care is associated with increased survival in adult trauma patients in Scotland. Emerg Med J. 2020;37:141–5. 

12. Lyons J, Gabbe BJ, Rawlinson D, Lockey D, Fry RJ, Akbari A, Lyons RA. Impact of a physician ? critical care practitioner pre-hospital service in Wales on trauma survival: a retrospective analysis of linked registry data. Anaesthesia. 2021; doi: 10.1111/anae.15457.

13. Bieler D, Franke A, Lefering R, Hentsch S, Willms A, Kulla M, et al. Does the presence of an emergency physician influence pre-hospital time, pre-hospital interventions and the mortality of severely injured patients? A matched-pair analysis based on the trauma registry of the German Trauma Society (TraumaRegister DGUR). Injury. 2017;48:32?40.

14. Yamamoto R, Cestero RF, Suzuki M, Funabiki T, Sasaki J. Resuscitative endovascular balloon occlusion of the aorta (REBOA) is associated with improved survival in severely injured patients: A propensity score matching analysis. Am J Surg. 2019 Dec;218(6):1162-1168. doi: 10.1016/j.amjsurg.2019.09.007. Epub 2019 Sep 13. PMID: 31540683.

15. Yamamoto R, Cestero RF, Muir MT, Jenkins DH, Eastridge BJ, Funabiki T, et al. Delays in Surgical Intervention and Temporary Hemostasis Using Resuscitative Endovascular Balloon Occlusion of the aorta (REBOA): Influence of Time to Operating Room on Mortality. Am J Surg. 2020;220:1485–91. 

16. Sadek S, Lockey DJ, Lendrum RA, Perkins Z, Price J, Davies GE. Resuscitative endovascular balloon occlusion of the aorta (REBOA) in the pre-hospital setting: An additional resuscitation option for uncontrolled catastrophic haemorrhage. Resuscitation. 2016;107:135–8.

17. Gamberini L, Coniglio C, Lupi C, Tartaglione M, Mazzoli CA, Baldazzi M, et al. Resuscitative endovascular occlusion of the aorta (REBOA) for refractory out of hospital cardiac arrest. An Utstein-based case series. Resuscitation. 2021;165:161–9. 

18. Taghavi S, Maher Z, Goldberg AJ, Chang G, Mendioloa M, Anderson C, et al. An Eastern Association for the Surgery of Trauma Multicenter Trial Examining Prehospital Procedures in Penetrating Trauma Patients. J Trauma Acute Care Surg. 2021; doi: 10.1097/TA.0000000000003151.

19. Dinh MM, Bein K, Roncal S, Byrne CM, Petchell J, Brennan J. Redefining the golden hour for severe head injury in an urban setting: the effect of prehospital arrival times on patient outcomes. Injury. 2013;44:606–10. 

20. Alarhayem AQ, Myers JG, Dent D, Liao L, Muir M, Mueller D, et al. Time is the enemy: Mortality in trauma patients with hemorrhage from torso injury occurs long before the "golden hour". Am J Surg. 2016;212:1101–5.

21. Brown JB, Rosengart MR, Forsythe RM, Reynolds BR, Gestring ML, Hallinan WM, et al. Not all prehospital time is equal: Influence of scene time on mortality. J Trauma Acute Care Surg. 2016;81:93–100.

---

## [Decision Letter · Decision Letter 1]

26 Oct 2021

Impact of emergency physician-staffed ambulances on preoperative time course and survival among injured patients requiring emergency surgery or transarterial embolization: A retrospective cohort study at a community emergency department in Japan

PONE-D-21-20967R1

Dear Dr. Ono,

We’re pleased to inform you that your manuscript has been judged scientifically suitable for publication and will be formally accepted for publication once it meets all outstanding technical requirements.

Kind regards,

Chun Chieh Yeh, M.D., Ph.D.

Academic Editor

PLOS ONE

Additional Editor Comments (optional):

Thanks for your revision. Our reviewers both endorsed your work.

Reviewers' comments:

Reviewer's Responses to Questions

**Comments to the Author**

1. If the authors have adequately addressed your comments raised in a previous round of review and you feel that this manuscript is now acceptable for publication, you may indicate that here to bypass the “Comments to the Author” section, enter your conflict of interest statement in the “Confidential to Editor” section, and submit your "Accept" recommendation.

Reviewer #1: All comments have been addressed

Reviewer #2: All comments have been addressed

2. Is the manuscript technically sound, and do the data support the conclusions?

Reviewer #1: Yes

Reviewer #2: Yes

3. Has the statistical analysis been performed appropriately and rigorously? 

Reviewer #1: Yes

Reviewer #2: Yes

4. Have the authors made all data underlying the findings in their manuscript fully available?

Reviewer #1: Yes

Reviewer #2: Yes

5. Is the manuscript presented in an intelligible fashion and written in standard English?

Reviewer #1: Yes

Reviewer #2: Yes

6. Review Comments to the Author

Reviewer #1: Thank you for your responses. I am satisfied with the revised version and congratulate the authors with this study.

Reviewer #2: (No Response)

7. PLOS authors have the option to publish the peer review history of their article (what does this mean?). If published, this will include your full peer review and any attached files.

Reviewer #1: No

Reviewer #2: No

---

## [Editor Report · Acceptance letter]

28 Oct 2021

PONE-D-21-20967R1 

Impact of emergency physician-staffed ambulances on preoperative time course and survival among injured patients requiring emergency surgery or transarterial embolization: A retrospective cohort study at a community emergency department in Japan 

Dear Dr. Ono:

I'm pleased to inform you that your manuscript has been deemed suitable for publication in PLOS ONE. Congratulations! Your manuscript is now with our production department. 

Kind regards, 

on behalf of

Dr. Chun Chieh Yeh 

Academic Editor

PLOS ONE